# Human Exposure to Bisphenols, Parabens, and Benzophenones, and Its Relationship with the Inflammatory Response: A Systematic Review

**DOI:** 10.3390/ijms24087325

**Published:** 2023-04-15

**Authors:** Francisco Manuel Peinado, Luz María Iribarne-Durán, Francisco Artacho-Cordón

**Affiliations:** 1Instituto de Investigación Biosanitaria de Granada (ibs.GRANADA), 18012 Granada, Spain; 2CIBER Epidemiology and Public Health (CIBERESP), 28029 Madrid, Spain; 3Radiology and Physical Medicine Department, University of Granada, 18016 Granada, Spain

**Keywords:** bisphenol, paraben, benzophenone, human, exposure, inflammation

## Abstract

Bisphenols, parabens (PBs), and benzophenones (BPs) are widely used environmental chemicals that have been linked to several adverse health effects due to their endocrine disrupting properties. However, the cellular pathways through which these chemicals lead to adverse outcomes in humans are still unclear, suggesting some evidence that inflammation might play a key role. Thus, the aim of this study was to summarize the current evidence on the relationship between human exposure to these chemicals and levels of inflammatory biomarkers. A systematic review of peer-reviewed original research studies published up to February 2023 was conducted using the MEDLINE, Web of Science, and Scopus databases. A total of 20 articles met the inclusion/exclusion criteria. Most of the reviewed studies reported significant associations between any of the selected chemicals (mainly bisphenol A) and some pro-inflammatory biomarkers (including C-reactive protein and interleukin 6, among others). Taken together, this systematic review has identified consistent positive associations between human exposure to some chemicals and levels of pro-inflammatory biomarkers, with very few studies exploring the associations between PBs and/or BPs and inflammation. Therefore, a larger number of studies are required to get a better understanding on the mechanisms of action underlying bisphenols, PBs, and BPs and the critical role that inflammation could play.

## 1. Introduction

During the past decades, there has been a growing public concern about the harmful effects that environmental phenols, including bisphenols, parabens (PBs), and benzophenones (BPs) could exert on human health [1,2,3,4,5,6].

Bisphenols are non-persistent phenolic compounds widely used in the synthesis of polycarbonate plastics and epoxy resins and are frequently found in the linings of canned and packaged food containers, thermal receipts, and dental sealants [7,8]. Bisphenol A (BPA) is the most studied congener and is one of the most produced chemicals in the world [9], reaching a global production volume of more than 5 million tons [10], and with an annual growth rate that reached 4.6% between 2013 and 2019 [11]. Moreover, data from biomonitoring studies indicate that BPA exposure is ubiquitous and widespread in the population, with BPA concentrations found in 90.0% of the general population in industrialized countries [12,13]. Due to the harmful effects inherent to exposure to BPA, some international government regulators have banned its use in baby bottles and cosmetics [14]. As an alternative to BPA, bisphenol analogs structurally similar to BPA began to be produced, such as bisphenol S, bisphenol F, bisphenol AF, tetrabromobisphenol, bisphenol A-glycidyl methacrylate, bisphenol A diglycidyl ether, and bisphenol F diglycidyl ether [15,16]. However, previous evidence has suggested that these analogs may be even more harmful than the original BPA in some situations [16]. The family of PBs includes alkyl esters of *p*-hydroxybenzoic acid, and such chemicals are used in a wide range of cosmetics and personal care products (PCPs) as well as in food packaging due to their antimicrobial and preservative properties [17,18,19,20,21]. The main congeners of PBs are methylparaben, ethylparaben, propylparaben, and butylparaben. BPs are aromatic ketones included in a wide variety of cosmetics, PCPs, and textiles due to their properties as UV filters [22,23], and include different congeners such as benzophenone 1, benzophenone 2, benzophenone 3, 4-hydroxybenzophenone, benzophenone 6, and benzophenone 8. As a result of these different uses, humans are widely exposed to these compounds through different pathways. While humans are mainly exposed to bisphenols through the diet [24,25], PBs and BPs are suspected to reach body compartments primarily through dermal absorption or consumption of packaged foodstuff [26,27]. Despite these compounds being rapidly metabolized and excreted by the body, the public concern regarding their potential health effects derives from the daily pattern of this exposure.

Indeed, previous studies have suggested that daily exposure to different bisphenol congeners might be associated with risks in women of miscarriage, endometriosis, polycystic ovary syndrome, thyroid disease, diabetes mellitus, obesity, cardiovascular disease, and metabolic syndrome [1,28,29,30,31,32]. In addition, PBs and BPs could also be the origin of adverse effects on human health, such as decreased body weight and height in children [33], decreased serum thyroid levels in humans [34], obesity [35], and gynecological disorders [36,37].

Nevertheless, despite the currently suspected harmful effects of these environmental phenols on human health, there are still several gaps of knowledge concerning their mechanisms of action. Currently, it is well known that bisphenols, PBs, and BPs have the ability to alter the homeostasis of the hormonal system due to their (anti-)estrogenic, (anti-)androgenic, and/or (anti-)thyroid actions [38,39,40] and therefore are considered endocrine-disrupting chemicals (EDCs). In addition, it has been postulated that inflammation might act as an alternative or complementary mechanism of action to the endocrine disruption hypothesis, given that they could promote an inflammatory milieu through activation of nuclear estrogen receptor (ER) α [41,42,43]. In this sense, previous evidence has reported the presence of estrogen-dependent nuclear receptors in promoter regions of genes related to the inflammatory response, such as ERα and ERβ [41,44,45,46], suggesting that the origin and development of an inflammatory response could be an indirect consequence of endocrine alterations promoted by these compounds with hormonal activity.

Inflammation is a regulatory mechanism for maintaining tissue homeostasis. It consists of a protective response of vascularized tissues to fight against a variety of challenges from the external environment, including those from infectious agents and tissue damage. It provides pathways for the rapid destruction of invading pathogens through the mobilization of immune cells across the vasculature and for the removal of damaged cells and tissues that may have been compromised in host defense [47]. A large number of biochemical reactions and mediators, such as cytokines, phagocytic leukocytes, antibodies, complement proteins, and intracellular adhesion molecules, among others, are involved in this complex process. Like most immune responses, the inflammatory phenomenon is tightly regulated, and a proper and precise balance between proinflammatory and anti-inflammatory immune responses is required to effectively eliminate infectious pathogens while limiting immune damage in the host [48]. The regulation of inflammatory responses is complex, involves many different cell types (immune, epithelial, endothelial, and mesenchymal cells) [41,47], and sometimes it may not be properly regulated. Misregulated inflammation can be initiated when the response among innate immune cells is inappropriate for the type of defense needed against the invader, the response is misdirected based on the location of the strange agent, the response is overproduced, and/or the response is not beneficially resolved for the host [41,47]. Deviations from tightly regulated inflammation present a significant health risk because unresolved inflammation can compromise tissue function and increase the risk of several chronic cardiovascular diseases and metabolic disorders [47].

In this sense, two previous systematic reviews have summarized the associations reported between exposure to different families of EDCs and inflammatory biomarkers [49,50]. However, the majority of EDCs explored were persistent organic pollutants, such as organochlorine pesticides (OCPs), and polychlorinated biphenyls. Considering non-persistent EDCs, only phthalates and BPA were explored [50] and currently there are no previous systematic reviews exploring the associations between other bisphenol congeners, PBs, or BPs and biomarkers of inflammation. Therefore, given (i) the ubiquity of these families of environmental phenols, and (ii) their possible adverse effects on health, there is a growing interest in the elucidation of potential mechanisms of action of these compounds. Thus, the aim of this study was to conduct a systematic review of published scientific evidence on associations between human exposure to bisphenols, PBs, and BPs and levels of inflammatory biomarkers.

## 2. Materials and Methods

This systematic review was conducted according to the Preferred Reporting Items for Systematic Reviews and Meta Analyses (PRISMA) statement [51].

### 2.1. Data Sources and Search Strategy

The databases MEDLINE (through the PubMed search engine), Web of Science (WoS), and Scopus were used to search for published studies reporting associations between human exposure to bisphenols, PBs, and BPs and levels of inflammatory biomarkers. The last search was performed on 1 February 2023. The detailed search strategy is displayed in Appendix A.

Our objective was to answer the question: “Is there evidence on associations between human exposure to bisphenols, PBs and BPs, and levels of inflammatory biomarkers?” We developed a PECO statement (Participants, Exposure, Comparator, and Outcomes) [52], which is used as an aid to developing an answerable question. Our PECO statement included the following:Participants: Humans.Exposure: Bisphenols [bisphenol A (BPA), bisphenol S (BPS), bisphenol F (BPF), bisphenol A-glycidyl methacrylate (BisGMA), bisphenol A diglycidyl ether (BADGE) and bisphenol F diglycidyl ether (BFDGE)], PBs [methylparaben (MeP), ethylparaben (EtP), propylparaben (PrP) and butylparaben (BuP)] and BPs (BP1-12).Comparators: Not applicable.Outcomes: Inflammatory biomarkers (cytokines, intracellular adhesion molecules, humoral mediators, C-reactive protein, inflammatory milieu, phagocytic leukocytes, antibodies, complement proteins, receptor activator of nuclear factor-kappa B, prostaglandin-endoperoxide synthases).

### 2.2. Study Selection and Data Extraction

Review inclusion criteria were: original scientific article; publication in English or Spanish; and the reporting of data on (i) the associations between human exposure to bisphenols, PBs, and BPs, and (ii) levels of inflammatory biomarkers. Exclusion criteria were: systematic and narrative reviews, case reports, conferences, meeting abstracts, and editorials; in vitro and in vivo studies.

Two researchers (LMID and FMP) independently conducted this systematic review. Firstly, the titles/abstracts of retrieved articles were screened, and duplicates and those not meeting the inclusion criteria were excluded. From the initially selected articles, the full text was reviewed and those that did not meet the inclusion criteria were then excluded. In case of discrepancy between reviewers, a third external reviewer (FAC) participated to make a decision about the inclusion or exclusion of the article at any step of the screening. The following data were collected from each article: (1) country; (2) type of study; (3) sample collection period of the exposure biomarker; (4) sample collection period of the inflammation biomarker; (5) sample size; (6) health condition; (7) gender; (8) age; (9) exposure (family and congeners) biomarkers; (10) inflammation biomarkers; (11) biological matrix; (12) chemical and biological quantification methodology; (13) extraction volume; (14) frequencies of detection (FD); (15) limit of detection of exposure biomarker; (16) units; (17) concentrations (arithmetic means, geometric means or percentile 50); (18) quality; (19) risk of bias; (20) statistical test; (21) magnitude of the reported associations, and (22) *p*-values of such associations. It is worth mentioning that in case of a variety of statistics reported to summarize EDC and/or inflammation biomarker concentrations, the median value was prioritized. Moreover, we have preserved units of measurements in tables, although we have appropriately unified them in order to make comparisons between studies.

### 2.3. Assessment of Reporting Quality and Risk of Bias

The reporting quality of the epidemiological studies was assessed using the Strengthening the Reporting of Observational studies in Epidemiology (STROBE) checklist [53]. This checklist consists of six blocks and a total of 23 items: (1) title and summary (2 items), (2) introduction (2 items), (3) method (9 items), (4) results (5 items), (5) discussion (4 items), and (6) other information (1 item). The reporting quality of articles was categorized according to Alvarenga, et al. [54] as high (≥16 items checked), medium (8–15 items), or low (<8 items) (Table 1).

The risk of bias was estimated by using a modified version of the Risk of Bias in Non-randomized Studies of Exposures (ROBINS-E) tool [55]. This tool comprises seven domains for the overall assessment of the risk of bias, including bias due to confounding; bias in selecting participants in the study; bias in exposure classification; bias due to departures from intended exposures; bias due to missing data; bias in outcome measurement; and bias in the selection of reported results. Each study was classified as “low”, “some concerns”, “high”, or “very high” risk of bias after evaluating each domain.

The reporting quality and the risk of bias assessment were performed by two reviewers (FMP and LMID). Any disagreement was resolved through a consensus discussion with the involvement of a third reviewer (FAC).

**Table 1 ijms-24-07325-t001:** General characteristics of the studies included in this systematic review.

Article Number	Reference	Country	Study Design	Period of Sample Collection		Sample Size	Reporting Quality *
				For Exposure Assessment	For Outcome Assessment		
1	Ashley-Martin et al., 2015 [56]	Canada	Cohort	2008–2011	2008–2011	1258	High
2	Aung et al., 2019 [57]	USA	Cohort	2006–2008	2006–2008	482 (1628 samples)	High
3	Choi et al., 2017 [58]	South Korea	Cohort	2013	2013	200	High
4	Ferguson et al., 2016 [59]	USA	Case-control	2006–2008	2006–2008	482 (1695 samples)	High
5	Haq et al., 2020 [60]	Pakistan	Cross-sectional	N.R.	N.R.	400	High
6	Huang et al., 2017 [61]	Taiwan	Cohort	2014–2016	2014–2016	230	High
7	Jain et al., 2020 [62]	India	Cross-sectional	N.R.	N.R.	300	Medium
8	Kelley et al., 2019 [63]	USA	Cohort	2012–2015	2012–2015	56	High
9	Lang et al., 2008 [64]	USA	Cross-sectional	2003–2004	2003–2004	1455	High
10	Liang et al., 2020 [65]	China	Cross-sectional	2015–2016	2015–2016	111	High
11	Linares et al., 2021 [66]	Spain	Prospective Observational	N.R.	N.R.	200	Medium
12	Mohsen et al., 2018 [67]	Egypt	Cross-sectional	N.R.	N.R.	167	High
13	Nalbantoğlu et al., 2021 [68]	Turkey	Case-control	2018	2018	280	High
14	Qu et al., 2022 [69]	China	Case-control	2018–2020	2018–2020	290	High
15	Savastano et al., 2015 [70]	Italy	Cross-sectional	N.R.	N.R.	76	High
16	Šimková et al., 2020 [71]	Czech Republic	Case-control	N.R.	N.R.	39	Medium
17	Song et al., 2017 [72]	South Korea	Cross-sectional	2008–2012	2008–2012	612 (1141 samples)	Medium
18	Tsen et al., 2021 [73]	Taiwan	Cross-sectional	N.R.	N.R.	90	Medium
19	Watkins et al., 2015 [43]	Puerto Rico	Cohort	2010–2012	2010–2012	106 (238 samples)	High
20	Yang et al., 2009 [74]	Korea	Cross-sectional	2005	2005	485	High

USA: United States of America; N.R.: not reported. * STROBE checklist items < 8: “low quality”; 8–15: “medium quality”; ≥16: “high quality”.

## 3. Results

Figure 1 depicts the PRISMA flow chart of articles through the study. A total of 3508 articles (1524 scientific papers were identified from MEDLINE, 469 from WoS, and 1515 from the Scopus database) were identified after applying the search strategy, from which 1182 were excluded for being duplicates. Then, titles and abstracts of the remaining 2326 articles were reviewed, and 2304 were excluded because no bisphenols, PBs or BPs were measured, no inflammation parameters were quantified, no Spanish or English language was used or they were review articles, conference papers, notes, book chapters, or conference abstracts. After full-text review of the remaining 22 articles, another two were excluded because they did not measure bisphenols, PBs, BPs or inflammation parameters or they were no epidemiological studies. Finally, 20 articles were selected for this review.

### 3.1. Characteristics of Studies

Table 1 exhibits the main characteristics of the 20 studies [43,56,57,58,59,60,61,62,63,64,65,66,67,68,69,70,71,72,73,74]. All included studies were published in the last 15 years (2008–2023). Nine out of 20 studies (45.0%) had a cross-sectional design, while six (30.0%) and four (20.0%) were cohort and case-control studies, respectively. A final study had a prospective observational design. A total of 13 out of 20 studies (65.0%) reported the period of time when biological samples for exposure/outcome assessment were collected. Given that they reported similar periods of biological samples collection for exposure and outcome assessment, most of studies were considered to have a cross-sectional design (for the purpose of this systematic review) despite some of them declaring a different study design. The sample size of the studies ranged from 39 to 1455 participants, with a pooled sample size of 7319 participants (10,339 samples). Studies were carried out in Asian (45.0%), American (30.0%) and European countries (20.0%). The reporting quality was classified as high in 15 studies (75.0%), while five studies (25.0%) had medium reporting quality [62,66,71,72,73]. The risk of bias was classified as very high in two studies [58,71], high in six studies [59,62,65,66,70,72], with some concerns in three studies [56,61,68], and low in nine studies [43,57,60,62,63,67,69,73,74] (Figure 2). Taken together, the higher concerns were due to confounding, the measurement of the exposure, and missing data, as shown in Figure 3.

Characteristics of study participants in the included studies are depicted in Appendix A. Two studies (10.0%) [67,68] were focused on children (boys and girls) and 18 on adults [one in men (5.0%), eight in women (40%; six of them during pregnancy) and nine (45.0%) included men and women]. The majority of studies included healthy population exclusively (*n* = 13, 65.0%), five studies combined healthy and pathologic patients (people with diabetes, allergic rhinitis, rheumatoid arthritis, and polycystic ovary syndrome), one study was focused on patients with Crohn’s disease and another study on women with unexplained recurrent spontaneous abortion.

### 3.2. Exposure of Bisphenols, PBs, and BPs

Table 2 provides an overview of the main methodological characteristics related to the exposure assessment for bisphenols, PBs, and BPs in the selected studies. Most of the studies used urine (13, 65.0%), while seven studies (35.0%) assessed the exposure in serum/plasma/blood samples. Regarding the families of chemicals assessed, 14 out 20 studies (70.0%) were focused on the exposure to bisphenols (mainly BPA), four (20.0%) on the exposure of the three families (bisphenols, PBs, and BPs) [43,57,63,66], one on bisphenols and PBs [71], and another only assessed PB concentrations [69]. The most frequently detected compounds were BPA [Frequency of detection (FD): 76.0–100%], MeP (FD: 97.0–100%), and BP-3 (FD: 99.7–100%). Concentrations of each studied chemical are also summarized in Table 2. Bisphenols, PBs, and BPs showed a median concentration ranging from <Limit of detection (LOD)-2.7 ng/mL, <LOD-186.0 ng/mL, and 34.5–42.6 ng/mL, respectively. BPA, MeP, and BP-3 were the most detected congeners of each of the three EDCs families explored. Other characteristics of the exposure assessment are summarized in Appendix A, including details on quantification methodology, limits of detection, and volume of sample used for the determination of the exposure in each study.

### 3.3. Inflammation Assessment

Table 3 provides an overview of the evaluation of the inflammation biomarkers assessed in the selected studies. All the studies quantified inflammation biomarkers in blood-related samples, half of them in the serum fraction of the blood (50.0%). Regarding the biomarkers assessed, C-reactive protein (CRP) was evaluated in 12 out 20 studies (60.0%); of these, half of them had CRP as the exclusive inflammation biomarker assessed. Interleukin 6 (IL-6) was the most common interleukin assessed (*n* = 11, 55.0%), followed by interleukin 10 (IL-10) (*n* = 6, 30.0%). Tumor necrosis factor-α (TNF-α) was quantified in nine studies (45.0%). The concentrations reported in each study are also summarized in Table 3. Briefly, CRP levels showed a mean concentration ranging from 2.6–678.0 ng/mL, and IL-6, IL-10, and TNF-α levels showed a median concentration range of <0.1–770.0 ng/mL, <0.1–0.2 ng/mL, and <0.1–1900.0 ng/mL, respectively. Other characteristics of outcome assessment are depicted in Appendix A, including the quantification methodology used and the frequency of detection of each biomarker.

### 3.4. Association between Exposure to Bisphenols, PBs and BPs, and Inflammation Biomarkers

As shown in Table 4, positive associations were identified between exposure to all bisphenols, PB and BP congeners, and levels of some inflammatory biomarkers. The great majority of the studies assessing BPA (12 out 18 studies, 66.7%) reported BPA-related increased levels of some proinflammatory cytokines, including CRP, monocyte chemoattractant protein 1 (MCP-1), interferon-γ (IFN-γ), interleukin 23 (IL-23), interleukin 17A (IL-17A), IL-6, TNF-α, alanine aminotransferase (ALT), aspartate aminotransferase (AST), and γ-glutamyl transferase (γ –GTP). Regarding PBs, only half of the studies assessing the influence of PBs on inflammation biomarkers (three out of six studies) reported significant associations with any inflammation biomarker. However, despite the fact that elevated levels of CRP and IL-6 were found to be related to MeP and increased CRP levels were related to PrP and BuP, Aung, et al. [57] and Watkins, et al. [43] also found an inverse association between EtP exposure and interlukin 1β (IL-1β) levels and between BuP exposure and CRP levels, respectively. Finally, two out four studies addressing exposure to BPs [43,57] identified a significant inverse association with TNF-α and CRP, respectively (Table 4).

## 4. Discussion

To date, this is the first systematic review gathering epidemiological studies exploring associations between exposure to bisphenols, PBs and BPs, and levels of inflammatory biomarkers. Most of the 20 included studies focused on the associations between BPA and inflammation, while the relationship between PBs/BPs and inflammation was only addressed in few studies (*n* = 6 and *n* = 4, respectively). Moreover, although most studies were focused on well-known inflammatory biomarkers such as CRP, IL-6, IL-10, IL-1α or IL-1β, more than 30 biomarkers of inflammation have been addressed (CRP, IL-1α, IL-1ra, IL-1β, IL-2, IL-4, IL-5, IL-6, IL-7, IL-8, IL-9 IL-10, IL-12, IL-13, IL-15, IL-17a, IL-23, IL-33, TNF-α, TGF-β, IFN-γ, GM-CSF, MCP-1, MCP-3, MIP1-a, MIP1-b, VEGF, FGF-basic, eotaxin, PDGF-BB, RANTES, ALT, AST, and γ-GTP). More than a half of the studies included in this review (*n* = 13, 65.0%) reported significant associations between any of the target EDCs included in this review and different inflammation parameters [43,56,57,58,59,63,65,66,68,69,70,72,74]. However, one-quarter of the selected studies were classified as medium reporting quality, and a certain degree of risk of bias was observed in more than half of the selected studies.

Briefly, BPA exposure was positively associated with a variety of pro-inflammatory biomarkers (CRP [58,60,64,74], IL-6 [59,60,70], IL-4 [68], IL-17A [66], IL-23 [66], IL-33 [56], MCP-1 [63], ALT [72], and γ-GTP [72]). Moreover, both MeP and PrP were associated with higher CRP levels [69], while MeP concentrations were also related to increased serum levels of IL-6 [57]. These results are in line with previous in vitro and in vivo studies reporting consistent positive associations between inflammation and exposure to bisphenols [75,76,77,78] and parabens [79]. Thus, the epidemiological findings summarized in this review, together with the in vitro and in vivo evidence, strongly support our hypothesis on the relationship between exposure to bisphenols, PBs, and BPs, and the development of an inflammatory response. Moreover, the underlying mechanisms explaining this association between exposure to these chemicals and inflammation might be related to the xenoestrogenic activity exhibited by bisphenols, PBs, and BPs. In this regard, it has been reported that low estrogenic activity promotes the production of type I interferon and pro-inflammatory cytokines [80]. Since these chemicals have nearly 1000-fold weaker affinity for ERs than estradiol, they can bind with ERs more actively when estrogen levels are low, which would in turn trigger physiological responses associated with inflammation [74].

However, a few inverse associations were also observed in previous studies. For instance, Watkins, et al. [43] found that exposure to BuP and BP-3 was related to lower CRP in pregnant women. Similarly, despite the fact that Aung, et al. [57] reported some positive associations between exposure and pro-inflammatory biomarkers, they also observed that EtP exposure during pregnancy was related to lower IL-1β levels and BP-3 to reduced TNF-α production. Nevertheless, sensitivity analyses of interaction terms between individual exposure analytes and study visits indicated that the association between EtP and IL-1β differed across study visits, becoming positive by visit 4 (33–38 gestational weeks). Moreover, a previous in vivo study revealed that inhibition of edema, an anti-inflammatory effect, was associated with topical BP-3 application [81].

To date, there is currently growing concern about the effects that human exposure to bisphenols, PBs, and BPs may have on health, and the adverse effects on human health of these chemicals are suggested to be related to their disruption of the endocrine system due to xenoestrogenic, xenoandrogenic and xenothyroid activities [38,39,40,82]. However, the exact mechanisms of action are not fully elucidated, with some evidence suggesting that these chemicals could exert adverse effects on human health though the perturbation of the oxidative microenvironment via ER-dependent pathways [43,83]. It is suspected that exposure to bisphenols, PBs, and BPs could have an immunotoxic effect, producing alterations in the immune system and deregulating inflammatory pathways through interactions with immune cells and peripheral tissues [84,85,86]. In this sense, a deregulation of the inflammation pathway is becoming increasingly important, as there is a growing amount of evidence reporting a relationship between disturbances in the inflammatory milieu and a multitude of allergic, autoimmune and reproductive diseases, as well as obesity, metabolic syndrome, and cancer, [47,87]. In fact, some of the studies of this review (*n* = 6, 20%) included in their study population participants with different diseases, such as diabetes [60,62], Crohn’s disease [66], allergic rhinitis [68], rheumatoid arthritis [69], and polycystic ovary syndrome (PCOS) [71]. Haq, et al. [60] and Jain, et al. [62] reported higher urinary BPA levels in diabetic participants compared with non-diabetics, and urinary BPA levels were correlated with elevated levels of CRP [60], TNF-α, IL-6, and IL-1α [62]. Linares, et al. [66] reported higher BPA levels in participants with active Crohn’s disease compared to participants with this disease in remission, along with positive correlations between BPA concentrations and IL-23 and IL-17a levels. Nalbantoğlu, et al. [68] also evidenced an association between BPA concentrations and allergic rhinitis in children, with increased levels of both BPA and IL-4 in more severe stages of the disease. Qu, et al. [69] reported significant associations among MeP and PrP exposure, increased CRP levels, and risk of rheumatoid arthritis, and Šimková, et al. [71] showed higher levels of BPA and IL-6, VEGF, and PDGFbb in PCOS women compared to controls. Taken together, these results suggest that a deregulated inflammatory response could be the nexus between the association between bisphenols, PBs, and BPs and the development and/or progression of diseases related to an altered immune system.

It is also important to highlight that humans are exposed to several toxicants and complex mixtures of EDCs and that their effects are difficult to predict given the possible synergistic, additive or antagonistic actions between chemical residues [88], while suspecting that they may be acting through immunological mechanisms [87]. Therefore, studies exploring associations between complex mixtures of bisphenols, PBs, and BPs with inflammatory biomarker levels acquire high importance. In this review, only one study [63] explored the combined effect of exposure to multiple chemicals. However, the inclusion of different families of chemicals to those considered for this systematic review (heavy metals, phthalates and other environmental phenols) hampered the elucidation of the specific contribution of bisphenols, PBs, and BPs to the inflammatory response. Therefore, future studies are required to address this combined effect on the inflammatory response. In addition, the great majority of studies (16, 80.0%) considered spot urine/blood samples for exposure assessment. Given that bisphenols, PBs, and BPs have a very short urinary elimination half-life in the human body [40,89], and thus spot samples may not be representative of the overall exposure of an individual, future studies considering pooled samples for exposure assessment (i.e., collecting 24-h urine/blood samples or repeated measurements) would show a more realistic scenario in relation to human exposure. In fact, this approach might yield stronger and more consistent associations between exposure and inflammatory biomarkers.

Moreover, it is plausible that there is a differential interference of exposure to these chemicals on inflammatory milieu according to the specific characteristics of study participants. For instance, inflammatory disruption might be more pronounced in vulnerable populations. Previous evidence suggests that EDCs could have a greater effect on health when human exposure occurs during critical periods of individual development, such as pregnancy, lactation, childhood or puberty [90,91]. In this sense, less than half of the studies included in this review (*n* = 8, 40.0%) considered these critical periods, of which six explored associations between exposure and inflammation in pregnant women [43,56,57,59,61,63] and two in children [67,68]. The results of these studies showed discrepant associations between exposure and inflammation, requiring a larger number of studies to be able to establish reliable conclusions between exposure and inflammation when considering critical windows of vulnerability. Furthermore, previous evidence has suggested the existence of gender differences related to inflammatory diseases [92]. For this reason, the inclusion of a study population gathering data from both genders becomes more important. In this review, almost half of the studies (*n* = 9, 45.0%) did not study gender differences [43,56,57,59,61,63,65,70,71], requiring further research in this regard.

Thus, the variability on the characteristics of study participants prevented the identification of subgroups of people with a higher risk for dysregulation of the inflammatory response with the exposure to bisphenols, PBs, and BPs. Additionally, the great methodological heterogeneity in terms of assessed inflammatory biomarkers and statistical analyses hampered the performance of a meta-analysis. In addition, taken together, the great heterogeneity in terms of study population (i.e., healthy adult subjects, healthy boys and girls, infant allergic rhinitis patients, arthritis rheumatoid patients, women diagnosed with PCOS, and diabetic and Crohn’s patients, etc.), the applied quantification methodology for both EDC (i.e., gas chromatography-mass spectrometry (GC-MS/MS), liquid chromatography-mass spectrometry (LC-MS/MS), isotope dilution-liquid chromatography-tandem mass spectrometry (ID-LC-MS/MS), enzyme-linked immunosorbent assay (ELISA), etc.) and inflammation assessment (i.e., ELISA, immunoturbidimetry, high performance liquid chromatography (HPLC), etc.), and the use of different biological matrices for exposure assessment (urine and serum) could explain, at least in part, the wide ranges reported in EDC and inflammatory biomarker levels shown in the different studies included in this review. Finally, concerns related to reporting quality and risk of bias were identified in one-quarter and a half of included studies, respectively. Thus, well-conducted studies are needed in the close future in order to obtain a more realistic overview on the contribution of these families of chemicals, to which humans are daily exposed, on the dysregulation of the inflammatory response.

Considering the limitations of this systematic review, the selection of the studies was based on the implementation of the search strategy in only three public databases. However, these databases are considered among the most relevant in the field of human health, and only a small number of specific publications that are only available in other databases could have been lost. On the other hand, only epidemiological studies were included in this review. This could limit the number of quantified inflammation biomarkers that could be included in other types of studies (in vivo and in vitro). Finally, although only three families of environmental phenols with endocrine-disrupting properties were selected for investigation, these include the phenolic EDCs in widest daily use, and a systematic review has already been carried out on the influence of organochlorine pesticides and PCBs on inflammation biomarkers [49].

In conclusion, this systematic review summarizes the current evidence on the association between human exposure to bisphenols, PBs, and BPs, and alterations in the inflammatory milieu. Despite some concerns related to reporting quality and risk of bias, selected studies showed consistent positive associations between human exposure to BPA and levels of some pro-inflammatory biomarkers, while very few studies explored associations between PBs and/or BPs and inflammation. Therefore, well-conducted studies in general, but also vulnerable, populations assessing exposure to both individual and mixtures of EDCs are required in the close future to clarify whether inflammation could act as a nexus between exposure to these EDCs and human health.

## Figures and Tables

**Figure 1 ijms-24-07325-f001:**
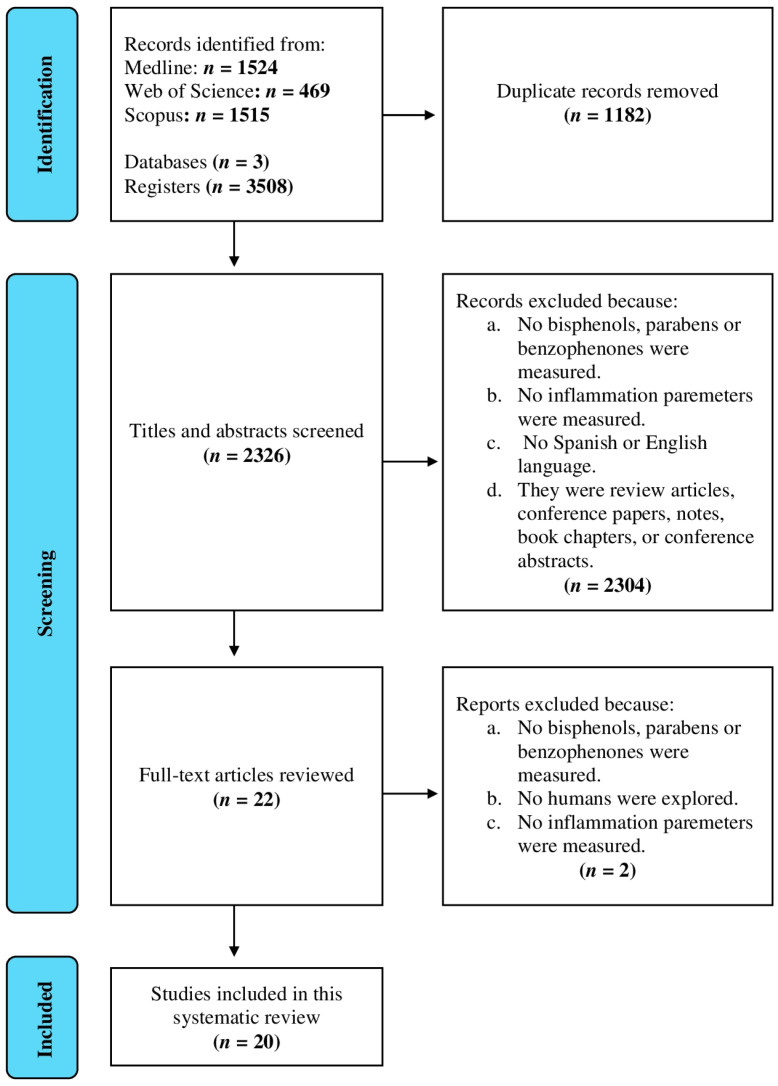
PRISMA flow chart for systematic review.

**Figure 2 ijms-24-07325-f002:**
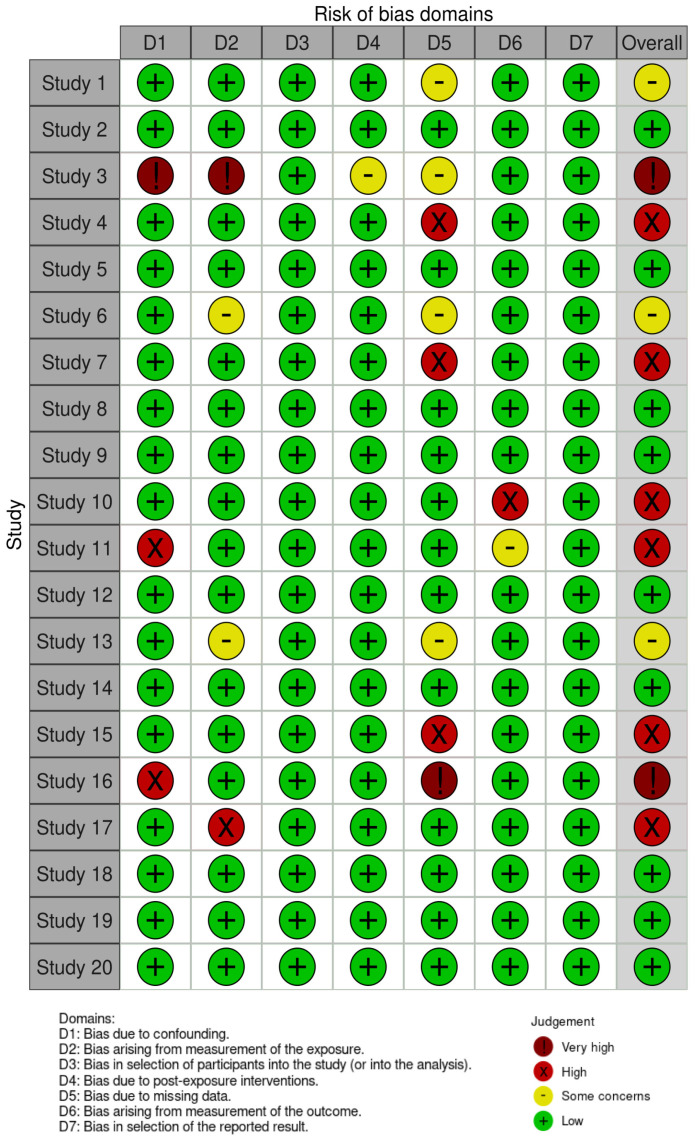
Assessment of the risk of bias of each study considering the ROBINS-E domains.

**Figure 3 ijms-24-07325-f003:**
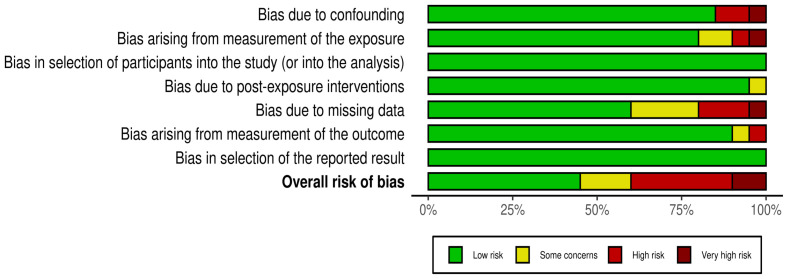
Assessment of the risk of bias of each ROBINS-E domain taking together all studies included in this review.

**Table 2 ijms-24-07325-t002:** Characteristics related to the exposure assessment to bisphenols, PBs, and BPs in the selected studies.

Article Number	Reference	EDC Family	Compounds	Matrix	Frequency of Detection (%)	Unit	Concentrations
1	Ashley-Martin et al., 2015 [56]	Bisphenols	BPA	Urine	86.6	µg/L	N.R.
2	Aung et al., 2019 [57]	Bisphenols	BPS	Urine	20.6	ng/mL	P50: 0.38
		PBs	MeP		99.9		P50: 186
			EtP		59.5		P50: 2.15
			PrP		99.0		P50: 45.60
			BuP		68.4		P50: 0.85
		BPs	BP-3		99.7		P50: 42.60
3	Choi et al., 2017 [58]	Bisphenols	BPA	Urine	N.R.	µg/L	N.R.
4	Ferguson et al., 2016 [59]	Bisphenols	BPA	Urine	83.4	ng/mL	GM: 1.32–1.38
5	Haq et al., 2020 [60]	Bisphenols	BPA	Urine	N.R.	ng/mL	Diabetic: 3.44 ± 1.82 *
							Healthy: 1.70 ± 0.43 *
6	Huang et al., 2017 [61]	Bisphenols	BPA	Urine	82.2	ng/mL	P50: 1.77
7	Jain et al., 2020 [62]	Bisphenols	BPA	Serum	N.R.	N.R.	N.R.
8	Kelley et al., 2019 [63]	Bisphenols	BPA	Urine	N.R.	N.R.	N.R.
			BPS				
			BPF				
		PBs	MeP				
			EtP				
			PrP				
			BuP				
		BPs	BP-3				
9	Lang et al., 2008 [64]	Bisphenols	BPA	Urine	N.R.	ng/mL	Men, weighted mean: 4.53
							Women, weighted mean: 4.66
10	Liang et al., 2020 [65]	Bisphenols	BPA	Urine	99.1	ng/mL	P50: 0.95
			BPS		41.4		P50: <LOD
11	Linares et al., 2021 [66]	Bisphenols	BPA	Serum	N.R.	µM	In remission: 5.57 ± 8.29 *
							Active disease: 11.98 ± 20.25 *
		PBs	MeP				In remission: 3.67 ± 5.72 *
							Active disease: 3.26 ± 5.50 *
			EtP				In remission: 0.90 ± 2.79 *
							Active disease: 0.31 ± 0.55 *
			PrP				In remission: 0.32 ± 0.82 *
							Active disease: 0.15 ± 0.25 *
			BuP				In remission: 0.07 ± 0.40 *
							Active disease: 0.04 ± 0.13 *
			BP-1				In remission: 0.10 ± 0.52 *
		BPs					Active disease: 0.05 ± 0.14 *
			BP-3				In remission: 0.21 ± 0.46 *
							Active disease: 0.03 ± 0.09 *
12	Mohsen et al., 2018 [67]	Bisphenols	BPA free	Urine	N.R.	ng/mL	P50 Boys: 0.20 P50 Girls: 0.21 P50 Boys: 0.25 P50 Girls: 0.33 P50 Boys: 0.60 P50 Girls: 0.67
			BPA conjugated				
			BPA total				
13	Nalbantoğlu et al., 2021 [68]	Bisphenols	BPA	Serum	N.R.	µg/L	Healthy: 445.38 ± 329.14 *
							Allergic rhinitis: 2225.83 ± 1321.75 *
14	Qu et al., 2022 [69]	PBs	MeP	Serum	Healthy: 97.0,	ng/mL	P50 Healthy: 2.60
					Rheumatoid arthritis: 100.0		P50 Rheumatoid arthritis: 4.70
			EtP		Healthy: 50.0,		P50 Healthy: 0.33
					Rheumatoid arthritis: 63.0		P50 Rheumatoid arthritis: 0.96
			PrP		Healthy: 53.0,		P50 Healthy: 0.49
					Rheumatoid arthritis: 71.0		P50 Rheumatoid arthritis: 0.74
			BuP		Healthy: 43.0,		P50 Healthy: <LOD
					Rheumatoid arthritis: 55.0		P50 Rheumatoid arthritis: 0.98
15	Savastano et al., 2015 [70]	Bisphenols	BPA	Plasma	N.R.	ng/mL	1.04 ± 0.77 *
16	Šimková et al., 2020 [71]	Bisphenols	BPA	Blood	Controls: 70.0	nM/L	P50 Controls: 0.13
					Normal weight PCOS: 100.0		P50 Normal weight PCOS: 0.28
					Obesity PCOS: 90.0		P50 Obesity PCOS: 0.13
			BPS		Controls: 25.0		P50 Controls: 0.00
					Normal weight PCOS: 33.0		P50 Normal weight PCOS: 0.00
					Obesity PCOS: 40.0		P50 Obesity PCOS: 0.00
			BPF		N.R.		N.R.
			BPAF		N.R.		N.R.
		PBs	MeP		N.R.		N.R.
			EtP		N.R.		N.R.
			PrP		N.R.		N.R.
			BuP		N.R.		N.R.
			BzP		N.R.		N.R.
			Total PBs		Controls: 30.0		P50 Controls: 0.00
					Normal weight PCOS: 56.0		P50 Normal weight PCOS: 0.49
					Obesity PCOS: 10.0		P50 Obesity PCOS: 0.00
17	Song et al., 2017 [72]	Bisphenols	BPA free	Urine	N.R.	µg/L	N.R.
			BPA conjugated				
18	Tsen et al., 2021 [73]	Bisphenols	BPA	Plasma	100.0	ng/mL	4.50 ± 2.00 *
19	Watkins et al., 2015 [43]	Bisphenols	BPA	Urine	98.7	ng/mL	P50: 2.67
		PBs	MeP		100.0		P50: 152.00
			PrP		100.0		P50: 45.40
			BuP		75.6		P50: 0.60
		BPs	BP-3		100.0		P50: 34.50
20	Yang et al., 2009 [74]	Bisphenols	BPA	Urine	76.0	µg/L	P50: 0.64

EDC: endocrine-disrupting chemical; PBs: parabens; BPs: benzophenones; BPA: bisphenol A; BPS: bisphenol S; BPF: bisphenol F; BPAF: bisphenol AF; BzP: benzylparaben; MeP: methylparaben; EtP: ethylparaben; PrP: propylparaben; BuP: butylparaben; BP-3: benzophenone 3; BP-1: benzophenone 1; PCOS: polycystic ovary syndrome; P50: percentile 50; GM: geometric mean; LOD: limit of detection; N.R.: Not reported. * Mean ± standard deviation.

**Table 3 ijms-24-07325-t003:** Characteristics related to the outcome assessment in the selected studies.

Article Number	Reference	Inflammation Parameter	Matrix	Unit	Concentrations
1	Ashley-Martin et al., 2015 [56]	IL-33	Cord blood	pg/mL	GM: 0.90
					GM: 0.90
2	Aung et al., 2019 [57]	CRP	Plasma	µg/mL	P50: 5.26
		IL-10		pg/mL	P50: 13.20
		IL-6			P50: 1.33
		TNF-α			P50: 2.99
		IL-1β			P50: 0.26
3	Choi et al., 2017 [58]	CRP	Serum	mg/L	0.63–4.57
4	Ferguson et al., 2016 [59]	CRP	Plasma	N.R.	N.R.
		IL-1β			
		IL-6			
		IL-10			
		TNF-α			
5	Haq et al., 2020 [60]	CRP	Blood	ng/mL	Diabetic BPA detected: Mean:10.63
					Diabetic BPA non detected: Mean: 7.50
					Non-diabetic BPA detected: Mean: 5.29
					Non-diabetic BPA non detected: Mean: 2.63
		IL-6		pg/mL	Diabetic BPA detected: Mean: 14.87
					Diabetic BPA non detected: Mean: 10.49
					Non-diabetic BPA detected: Mean: 4.62
					Non-diabetic BPA non detected: Mean: 2.75
6	Huang et al., 2017 [61]	CRP	Plasma and cord serum	µg/mL	P50 Plasma: 2.60
					P50 Cord serum: N.R.
		IL-6		pg/mL	P50 Plasma: 6.26
					P50 Cord serum: 3.70
		TNF-α		pg/mL	P50 Plasma: 3.65
					P50 Cord serum: 5.47
7	Jain et al., 2020 [62]	TNF-α	Serum	pg/mL	Diabetic population: 87.88 ± 26.77 *
					Control: 82.12 ± 27.45 *
		IL-6			Diabetic population: 103.89 ± 16.83 *
					Control: 101.76 ± 13.37 *
		IL-1α			Diabetic population: 62.42 ± 10.53 *
					Control: 60.15 ± 7.73 *
8	Kelley et al., 2019 [63]	GM-CSF	Blood and cord blood	pg/mL	N.R.
		IFN-γ			
		MCP-1			
		MCP-3			
		MIP-1α			
		MIP-1β			
		TNFα			
		VEGF			
		IL-1β			
		IL-6			
		IL-8			
		IL-17A			
9	Lang et al., 2008 [64]	CRP	Serum	N.R.	N.R.
10	Liang et al., 2020 [65]	IL-1β	Serum	ng/mL	P50: 0.08
		IL-2			P50: <LOD
		IL-4			P50: <LOD
		IL-6			P50: 0.70
		IL-8		µg/mL	P50: 0.06
		IL-10		ng/mL	P50: 0.17
		IL-12p70			P50: 0.01
		IL-13			P50: 0.24
		TNF-α			P50: 1.82
		TGF-β		µg/mL	P50: 17.13
		IFN-γ		ng/mL	P50: 5.54
11	Linares et al., 2021 [66]	IL-12	Serum	µg/mL	In remission: 38.60 ± 17.20 *
					Active disease: 42.50 ± 16.90 *
		IFN-γ			In remission: 21.10 ± 10.90 *
					Active disease: 26.13 ± 11.50 *
		IL-6			In remission: 28.90 ± 16.30 *
					Active disease: 27.70 ± 13.50 *
		IL-23			In remission: 12.60 ± 10.40 *
					Active disease: 16.50 ± 8.90 *
		IL-17A			In remission: 26.6 ± 11.60 *
					Active disease: 32.0 ± 16.60 *
12	Mohsen et al., 2018 [67]	CRP	Serum	ng/mL	Boys: 5.17 ± 7.01 *
					Girls: 4.13 ± 5.75 *
13	Nalbantoğlu et al., 2021 [68]	IL-4	Serum	µg/mL	Healthy: 14.28 ± 10.17 *
					Allergic rhinitis: 32.03 ± 26.45 *
		IL-13			Healthy: 9.09 ± 5.13 *
					Allergic rhinitis: 9.27 ± 5.44 *
		IFN-γ			Healthy: 5.12 ± 3.79 *
					Allergic rhinitis: 5.79 ± 4.13 *
14	Qu et al., 2022 [69]	CRP	Serum	mg/L	P25-P75 Controls: 1.60–2.40
					P25-P75 Cases: 4.30–55.30
15	Savastano et al., 2015 [70]	MCP1	Plasma	µg/mL	27.40 ± 23.50 *
		IL-6			P50: 0.77
		TNF-α			P50: 1.90
16	Šimková et al., 2020 [71]	FGF basic	Plasma	pg/mL	N.R.
		Eotaxin			N.R.
		GM-CSF			N.R.
		IFN-γ			P50 Controls: 19.90
					P50 Normal weight PCOS: 13.40
					P50 Obesity PCOS: 32.80
		IL-1β			N.R.
		IL-1ra			N.R.
		IL-2			P50 Controls: 18.00
					P50 Normal weight PCOS: 12.50
					P50 Obesity PCOS: 22.20
		IL-4			N.R.
		IL-5			N.R.
		IL-6			P50 Controls: 23.10
					P50 Normal weight PCOS: 56.70
					P50 Obesity PCOS: 82.10
		IL-7			N.R.
		IL-8			N.R.
		IL-9			N.R.
		IL-10			N.R.
		IL-12 (p70)			N.R.
		IL-13			P50 Controls: 7.38
					P50 Normal weight PCOS: 5.82
					P50 Obesity PCOS: 8.85
		IL-15			N.R.
		IL-17A			N.R.
		IP-10			N.R.
		MCP-1			N.R.
		MIP-1α			N.R.
		MIP-1β			N.R.
		PDGF-BB			P50 Controls: 216.00
					P50 Normal weight PCOS: 328.00
					P50 Obesity PCOS: 291.00
		RANTES			N.R.
		TNF-α			N.R.
		VEGF			P50 Controls: 459.00
					P50 Normal weight PCOS: 1028.00
					P50 Obesity PCOS: 1120.00
17	Song et al., 2017 [72]	CRP	Blood and serum	N.R.	N.R.
		IL-10			
		ALT			
		AST			
		γ-GTP			
18	Tsen et al., 2021 [73]	CRP	Plasma	ng/mL	678.00 ± 918.10 *
19	Watkins et al., 2015 [43]	CRP	Serum	N.R.	N.R.
		IL-1β			
		IL-6			
		IL-10			
		TNF-α			
20	Yang et al., 2009 [74]	CRP	Serum	mL/dL	Men: 0.08 ± 2.45 *
					Premenopausal women: 0.06 ± 3.63 *
					Postmenopausal women: 0.08 ± 3.00 *

IL: interleukin; CRP: C-reactive protein; TNF-α: tumor necrosis factor-α; GM-CSF: granulocyte macrophage colony-stimulating factor; IFN-γ: interferon-γ; MCP: monocyte chemoattractant protein; MIP: macrophage inflammatory protein; VEGF: vascular endothelial growth factor; TGF-β: transforming growth factor-β; FGF: fibroblast growth factor; PDGF-BB: platelet-derived growth factor-BB; RANTES: regulated upon activation, normal T-cell expressed and secreted; ALT: alanine aminotransferase, AST: aspartate aminotransferase; γ-GTP: γ-glutamyl transferase; GM: geometric mean; BPA: bisphenol A; LOD: limit of detection; P50: percentile 50; P25: percentile 25; P75: percentile 75; PCOS: polycystic ovary syndrome; N.R.: Not reported. * Mean ± standard deviation.

**Table 4 ijms-24-07325-t004:** Association between concentrations of bisphenols, parabens, and benzophenones and levels of inflammation biomarkers.

Article Number	Reference	Exposure-Inflammation Biomarkers	Statistical Test	Magnitude of the Association	*p*-Value
1	Ashley-Martin et al., 2015 [56]	BPA-IL-33	Bayesian hierarchical logistic regression models [OR (95% CI)]	1.00 (0.70–1.30)	0.050
2	Aung et al., 2019 [57]	MeP-CRP	Percent change (95% CI)	5.56 (−1.49–13.1)	0.130
		EtP-CRP		3.36 (−4.31–11.6)	0.400
		PrP-CRP		6.40 (−0.25–13.5)	0.060
		BuP-CRP		7.17 (−2.22–17.5)	0.140
		BP-3-CRP		0.79 (−6.44–8.59)	0.840
		MeP-IL-1β		−0.15 (−6.37–6.48)	0.960
		EtP-IL-1β		−7.70 (−14.1–−0.86)	0.030
		PrP-IL-1β		−2.36 (−8.01–3.63)	0.430
		BuP-IL-1β		−6.28 (−13.9–2.04)	0.130
		BP-3-IL-1β		1.05 (−5.83–8.43)	0.770
		MeP-IL-6		6.69 (0.02–13.8)	0.049
		EtP-IL-6		−4.20 (−10.9–2.95)	0.240
		PrP-IL-6		2.94 (−3.05–9.30)	0.340
		BuP-IL-6		−3.59 (−11.5–5.03)	0.400
		BP-3-IL-6		−1.60 (−8.32–5.61)	0.650
		MeP-IL-10		0.34 (−4.38–5.29)	0.890
		EtP-IL-10		−3.33 (−8.37–2.00)	0.220
		PrP-IL-10		−1.53 (−5.82–2.97)	0.500
		BuP-IL-10		0.80 (−5.42–7.44)	0.800
		BP-3-IL-10		−0.34 (−5.47–5.07)	0.900
		MeP-TNF-α		1.42 (−1.85–4.80)	0.400
		EtP-TNF-α		−3.14 (−6.61–0.46)	0.090
		PrP-TNF-α		−0.05 (−3.05–3.03)	0.970
		BuP-TNF-α		−0.42 (−4.66–4.00)	0.850
		BP-3-TNF-α		−3.69 (−7.09–−0.17)	0.040
3	Choi et al., 2017 [58]	BPA-CRP	Multiple logistic regression [OR (95% CI)]	2.85 (1.16–6.97)	0.022
4	Ferguson et al., 2016 [59]	BPA-CRP	Percent change (95% CI)	−1.64 (−8.63–5.88)	0.660
		BPA-IL-1β		3.36 (−3.41–10.60)	0.340
		BPA-IL-6		8.95 (1.81–16.60)	0.010
		BPA-IL-10		3.05 (−1.98–8.35)	0.240
		BPA-TNF-α		0.30 (−3.18–3.91)	0.860
5	Haq et al., 2020 [60]	BPA Detected-CRP	Two-tailed Student’s *t* test (mean ± SEM).	Diabetes: 10.63 ± 0.66	<0.05
		BPA Non-detected-CRP		Diabetic: 7.50 ± 1.51	
		BPA Detected-CRP		Non-diabetic: 5.29 ± 0.59	
		BPA Non-detected-CRP		Non-diabetic: 2.63 ± 0.34	
		BPA Detected-IL-6		Diabetes: 14.84 ± 0.63	<0.001
		BPA Non-detected-IL-6		Diabetic: 10.49 ± 0.76	
		BPA Detected-IL-6		Non-diabetic: 4.62 ± 0.37	
		BPA Non-detected-IL-6		Non-diabetic: 2.75 ± 0.21	
6	Huang et al., 2017 [61]	BPA-CRP (plasma)	Multivariate linear regression [β (SE)]	−0.06 (0.10)	0.570
		BPA-CRP (cord serum)		N.R.	N.R.
		BPA-Il-6 (plasma)		−0.82 (0.98)	0.400
		BPA-Il-6 (cord serum)		−0.74 (2.30)	0.750
		BPA-TNF-α (plasma)		−0.16 (0.32)	0.620
		BPA-TNF-α (cord serum)		−0.14 (0.26)	0.590
7	Jain et al., 2020 [62]	BPA-TNF-α (control population)	Spearman correlation (Sρ)	−0.07	0.940
		BPA-TNF-α (diabetes population)		−0.05	0.560
		BPA-IL-6 (control population)		−0.11	0.180
		BPA-IL−6 (diabetes population)		−0.04	0.660
		BPA-IL-1α (control population)		−0.05	0.510
		BPA-IL-1α (diabetes population)		0.04	0.660
8	Kelley et al., 2019 [63]	BuP-IL-6	Linear regression. Effect size (standard deviation)	−0.32 (0.11)	0.097
		BPA-MCP-1		0.82 (0.21)	0.019
		BPA, BPS, BPF, MeP, EtP, PrP, BuP, BP-3-GM-CSF, IFN-γ, MCP-1, MCP-3, MIP-1α, MIP-1β, TNFα, VEGF, IL-1β, IL-6, IL-8, and IL-17A		No significant correlations	N.R.
9	Lang et al., 2008 [64]	BPA-CRP	Multivariate linear regression [β (95% CI)]	0.09 (0.02–0.15)	0.020
10	Liang et al., 2020 [65]	BPA-IL-1β	Multivariate Linear regression [β (95% CI)]	0.31 (−0.48–1.10)	0.439
		BPA-IL-2		N.R.	N.R.
		BPA-IL-4		N.R.	N.R.
		BPA-IL-6		0.15 (−0.14–0.44)	0.314
		BPA-IL-8		0.06 (−0.31–0.46)	0.776
		BPA-IL-10		0.03 (−0.18–0.23)	0.801
		BPA-IL-12p70		−0.09 (−0.40–0.22)	0.573
		BPA-IL-13		0.26 (−0.17–0.69)	0.225
		BPA-TNF-α		0.00 (−0.16–0.16)	0.996
		BPA-TGF-β		−0.00 (−0.07–0.07)	0.981
		BPA-IFN-γ		0.18 (0.00–0.36)	0.045
		BPS-IL-1β		0.17 (−0.27–0.61)	0.433
		BPS-IL-2		N.R.	N.R.
		BPS-IL-4		N.R.	N.R.
		BPS-IL-6		0.03 (−0.13–0.19)	0.724
		BPS-IL-8		0.05 (−0.17–0.27)	0.644
		BPS-IL-10		0.06 (−0.06–0.17)	0.328
		BPS-IL-12p70		0.08 (−0.09–0.25)	0.340
		BPS-IL-13		0.07 (−0.17–0.31)	0.572
		BPS-TNF-α		−0.00 (−0.09–0.09)	0.984
		BPS-TGF-β		0.01 (−0.03–0.05)	0.658
		BPS-IFN-γ		−0.01 (−0.11–0.09)	0.890
11	Linares et al., 2021 [66]	BPA IL-23	Multivariate linear regression [β (95% CI)]	1.69 (1.60–1.77)	0.001
		BPA IL-17A		1.15 (1.00–1.29)	0.001
		MeP, EtP, PrP, BuP, BP-1, BP-3-IL-12, IFN-γ, IL-6, IL-23, IL-17A	N.R.	N.R.	N.R.
12	Mohsen et al., 2018 [67]	BPA-CRP	Spearman correlation coefficients (Sρ)	N.R.	N.R.
13	Nalbantoğlu et al., 2021 [68]	BPA-IL-4	Multivariate linear regression [β (95% CI)]	0.31 (3.47–7.40)	0.000
		BPA-IL-13		N.R.	N.R.
		BPA-IFN-γ		N.R.	N.R.
14	Qu et al., 2022 [69]	MeP-CRP	Multivariate linear regression [β (95% CI)]	0.15 (0.04–0.28)	<0.05
		EtP-CRP		0.23 (−0.11–0.56)	>0.05
		PrP-CRP		0.20 (0.10–0.32)	<0.05
		BuP-CRP		0.27 (−0.10–0.80)	>0.05
15	Savastano et al., 2015 [70]	BPA-MCP-1	Multivariate linear regression (β)	N.R.	N.R.
		BPA-IL-6		0.24	0.037
		BPA-TNF-α		N.R.	N.R.
16	Šimková et al., 2020 [71]	BPA, BPS, BPF, BPAF, Mep, EtP, PrP, BuP, BzP, total PBs-FGF basic, eotaxin, GM-CSF, IFN-γ, IL-1β, IL-1ra, IL-2, IL-4, IL-5, IL-6, IL-7, IL-8, IL-9, IL-10, IL-12 (p70), IL-13, IL-15, IL-17A, IP-10, MCP-1, MIP-1α, MIP-1β, PDGF-BB, RANTES, TNF-α, VEGF	Spearman correlation coefficients (Sρ)	No significant correlations	N.R.
17	Song et al., 2017 [72]	BPA-CRP	Linear mixed-effect model and a generalized additive mixed model (GAMM)	Positive non-linear association	0.081
		BPA-IL-10		Negative non-linear association	0.083
		BPA-ALT		Positive non-linear association	0.001
		BPA-AST		Positive non-linear association	0.056
		BPA-γ-GTP		Positive non-linear association	0.018
18	Tsen et al., 2021 [73]	BPA-CRP	Multiple logistic regression [OR (95% CI)]	1.82 (0.58–5.36)	0.283
19	Watkins et al., 2015 [43]	BPA-CRP	Percent change (95% CI)	5.10 (−7.47–19.40)	0.440
		BPA-IL-1β		4.65 (−7.91–18.90)	0.480
		BPA-IL-6		12.50 (−2.50–29.70)	0.110
		BPA-IL-10		−1.20 (−13.40–12.70)	0.850
		BPA-TNF-α		4.85 (−1.70–11.80)	0.150
		MeP-CRP		−6.75 (−19.00–7.38)	0.330
		MeP-IL-1β		−3.63 (−17.10–12.10)	0.630
		MeP-IL-6		4.90 (−11.20–23.90)	0.570
		MeP-IL-10		6.66 (−8.62–24.50)	0.410
		MeP-TNF-α		2.00 (−6.18–10.90)	0.640
		PrP-CRP		−13.60 (−25.80–0.50)	0.060
		PrP-IL-1β		−1.76 (−16.50–15.60)	0.830
		PrP-IL-6		3.70 (−13.40–24.20)	0.690
		PrP-IL-10		−0.09 (−15.40–18.00)	0.990
		PrP-TNF-α		−0.83 (−9.29–8.42)	0.850
		BuP-CRP		−17.50 (−30.30–−2.27)	0.030
		BuP-IL-1β		10.50 (−8.11–32.80)	0.290
		BuP-IL-6		15.80 (−5.47–41.70)	0.150
		BuP-IL-10		5.28 (−12.90–27.20)	0.590
		BuP-TNF-α		5.69 (−4.67–17.20)	0.290
		BP-3-CRP		−16.30 (−27.50–−3.42)	0.020
		BP-3-IL-1β		−0.75 (−15.10–16.10)	0.920
		BP-3-IL-6		−4.81 (−19.90–13.10)	0.570
		BP-3-IL-10		−3.88 (−18.10–12.80)	0.620
		BP-3-TNF-α		−2.13 (−10.30–6.77)	0.620
20	Yang et al., 2009 [74]	BPA-CRP	Multivariate linear regression (β)	Men: −0.02	0.418
		BPA-CRP		Premenopausal women: 0.09	0.268
		BPA-CRP		Postmenopausal women: 0.11	0.029

BPA: bisphenol A; BPS: bisphenol S; BPF: bisphenol F; BPAF: bisphenol AF; MeP: methylparaben; EtP: ethylparaben; PrP: propylparaben; BuP: butylparaben; BzP: benzylparaben; PBs: parabens; BP-1: benzophenone-1; BP-3: benzophenone 3; EDC: endocrine-disrupting chemical; IL: interleukin; CRP: C-reactive protein; TNF-α: tumor necrosis factor-α; EDC: endocrine-disrupting chemical; TGF-β: transforming growth factor-β; INF-γ: interferon-γ; MCP: monocyte chemoattractant protein; FGF: fibroblast growth factor; GM-CSF: granulocyte macrophage colony-stimulating factor; IP-10: interferon-γ-inducible protein 10; MIP: macrophage inflammatory protein; PDGF-BB: platelet-derived growth factor-BB; RANTES: regulated upon activation, normal T-cell expressed and secreted; VEGF: vascular endothelial growth factor; ALT: alanine aminotransferase, AST: aspartate aminotransferase; γ-GTP: γ-glutamyl transferase; OR: odds ratio; CI: confidence interval; SEM: standard error of mean, SE: standard error; Sρ: Spearman’s correlation coefficient; N.R.: Not reported.

## Data Availability

Not applicable.

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
