# Peer review of "Human Exposure to Bisphenols, Parabens, and Benzophenones, and Its Relationship with the Inflammatory Response: A Systematic Review"

_ijms, 2023, doi:10.3390/ijms24087325_

Round 1
Reviewer 1 Report
The manuscript entitled Human exposure to bisphenols, parabens and benzophenones, and its relationship with the inflammatory response. A systematic review is well designed and written. It draws attention to the problem of the anthropogenic chemicals, that can be harmful to human organisms. However, one should be aware that the topic discussed by the authors has been partly explored by other scientists recently. In 2022 Liu et al. published the article entitled The associations between endocrine disrupting chemicals and markers of inflammation and immune responses: A systematic review and meta-analysis. ( Ecotoxicol Environ Saf. 2022;234:113382. doi: 10.1016/j.ecoenv.2022.113382.). That article should be mentioned in the introduction.
I recommend the publication of the manuscript in International Journal of Molecular Sciences after addressing the following issues:
- inclusion of the aforementioned publication;
- P. 2, line 59 – change the phrase “central regulatory mechanism” into regulatory mechanism, as the “central” suggest central nervous system ;
- Authors should give the comments on the differences in the concentrations of studied chemicals (given in Table 2), as well as they should comment some of the differences in the determined biomarker concentrations (Table 3)
Also I feel that the information, that authors should discuss more thoroughly the problem of using blood serum to assess the exposure to quickly metabolized compounds, such as bisphenol A (e.g. described by Calafat et al. Misuse of blood serum to assess exposure to bisphenol A and phthalates. Breast Cancer Res 15, 403 (2013)).
Author Response
Reviewer #1:
The manuscript entitled Human exposure to bisphenols, parabens and benzophenones, and its relationship with the inflammatory response. A systematic review is well designed and written. It draws attention to the problem of the anthropogenic chemicals, that can be harmful to human organisms. However, one should be aware that the topic discussed by the authors has been partly explored by other scientists recently. In 2022 Liu et al. published the article entitled The associations between endocrine disrupting chemicals and markers of inflammation and immune responses: A systematic review and meta-analysis. (Ecotoxicol Environ Saf. 2022;234:113382. doi: 10.1016/j.ecoenv.2022.113382.). That article should be mentioned in the introduction.
Response: We are grateful for the positive evaluation by this reviewer, and for the interesting comments that have definitely improved the quality of this study. We agree with the reviewer that the systematic review carried out by Liu et al. (2022) partially explores the topic addressed in our review. Liu et al. (2022) explored the associations between markers of inflammation and endocrine-disrupting chemicals (EDCs) from two different groups: persistent EDCs [organochlorine pesticides (OCPs), and polychlorinated biphenyls (PCBs)], and non-persistent EDCs [phthalates and bisphenol A (BPA)]. Contrarily, our review is focused on bisphenols (BPA but also other bisphenol congeners such as bisphenol F, bisphenol S or bisphenol AF), parabens and benzophenones, all of them non-persistent EDCs typically used in plastics and daily use products such as food packaging, personal care products, and cosmetics. Consequently, the only chemical that has been addressed in both reviews is BPA. In this regard, it is worth to mention that our review did not focus only on a single bisphenol congener. For this reason, the search strategy shown in Supplementary Table 1 included the term “bisphenols” in general and specifically included different congeners, such as "bisphenol S", "bisphenol F", "bisphenol A-glycidyl methacrylate", “bisphenol A diglycidyl ether" and "bisphenol F diglycidyl ether". Therefore, despite the fact that both reviews explore BPA, we consider that the objectives of both systematic reviews are different, including different families of EDCs.
As requested, we have added a new paragraph in the introduction section and have cited the systematic review carried out by Liu et al. (2022). The included paragraph reads as follows:
“In this sense, two previous systematic reviews have summarized the associations reported between exposure to different families of EDCs and inflammatory biomarkers (Liu et al., 2022; Peinado et al., 2020). However, the majority of EDCs explored were persistent organic pollutants, such as organochlorine pesticides (OCPs), and polychlorinated biphenyls. Considering non-persistent EDCs, only phthalates and BPA were explored (Liu et al., 2022) and currently there are no previous systematic reviews exploring the associations between other bisphenol congeners, PBs, or BPs and biomarkers of inflammation.”
I recommend the publication of the manuscript in International Journal of Molecular Sciences after addressing the following issues:
- inclusion of the aforementioned publication;
Response: As mentioned above, we have included the aforementioned publication (Liu et al., 2022) in the introduction section.
- P. 2, line 59 – change the phrase “central regulatory mechanism” into regulatory mechanism, as the “central” suggest central nervous system ;
Response: As requested, we have modified the phrase "central regulatory mechanism" into "regulatory mechanism".
- Authors should give the comments on the differences in the concentrations of studied chemicals (given in Table 2), as well as they should comment some of the differences in the determined biomarker concentrations (Table 3)
Response: As requested, information regarding the differences in the concentrations of the chemicals studied and the levels of inflammatory biomarkers has been summarized in the results section. We have detected a wide heterogeneity in terms of the statistics used to summarize EDC and/or inflammation biomarkers concentrations along studies. For instance, some studies reported medians and others arithmetic means, being inappropriate to compare these results. In this regard, in case of a variety of statistics reported, we have prioritized median values given that it is more appropriate in cases of non-parametric distributions. This information has been clarified in the method section of the revised version of the manuscript, that now reads:
“It is worth to mention that in case of a variety of statistics reported to summarize EDC and/or inflammation biomarker concentrations, median value was prioritized. Moreover, we have preserved units of measurements in tables, although we have appropriately unified them in order to make comparisons between studies.”
For that reason, between-studies comparisons have been accomplished for those reporting median instead of arithmetic mean values. The exception was CRP, given that there was a higher number of studies reporting arithmetic means but not median values of this inflammation biomarker. Therefore, the median concentration of both endocrine disruptors and the levels of IL-6, IL-10 and TNF-α, and the mean CRP concentration were considered. The revised paragraph now reads as follows:
“Bisphenols, PBs, and BPs showed a median concentration ranging from <LOD-2.7 ng/mL, <LOD-186.0 ng/mL, and 34.5-42.6 ng/mL, respectively. BPA, MeP and BP-3 were the most detected congeners of each of the three EDCs families explored.”
“Briefly, CRP levels showed a mean concentration ranging from 2.6-678.0 ng/mL, and IL-6, IL-10, and TNF-α levels showed a median concentration range of <0.1-770.0 ng/mL, <0.1-0.2 ng/mL and <0.1-1900.0 ng/mL, respectively.”
In addition, we have added new information in the discussion section regarding the differences found in the concentrations of exposure to EDCs and levels of inflammatory biomarkers, which now reads as follows:
“In addition, taken together, the great heterogeneity in terms of study population (i.e. healthy adult subjects, healthy boys and girls, infant allergic rhinitis patients, arthritis rheumatoid patients, women diagnosed with PCOS, and diabetic and Crohn’s patients…), the applied methodological for both EDC (i.e. gas chromatography-mass spectrometry (GC-MS/MS), liquid chromatography-mass spectrometry (LC-MS/MS), isotope dilution-liquid chromatography-tandem mass spectrometry (ID-LC-MS/MS), enzyme-linked immunosorbent assay (ELISA)…) and inflammation assessment (i.e. ELISA, immunoturbidimetry, high performance liquid chromatography (HPLC)…), and the use of different biological matrices for exposure assessment (urine and serum) could explain, at least in part, the wide ranges reported in EDC and inflammatory biomarker levels shown in the different studies included in this review.”
Also I feel that the information, that authors should discuss more thoroughly the problem of using blood serum to assess the exposure to quickly metabolized compounds, such as bisphenol A (e.g. described by Calafat et al. Misuse of blood serum to assess exposure to bisphenol A and phthalates. Breast Cancer Res 15, 403 (2013)).
Response: We totally agree with the reviewer that blood serum is not the most appropriate biological matrix for the assessment of the exposure to bisphenols, parabens, and benzophenones. These compounds are rapidly metabolized and the levels of their polar, hydrophilic metabolites in blood could be several orders of magnitude lower than in urine. Urine is considered the best matrix for the determination of the concentrations of these families of EDCs, and only 65% of the studies included in this review (n=13) quantified the levels of EDCs in this matrix, while 35% (n=7) of the studies quantified the concentrations of EDCs in serum/plasma/blood samples. Given that the use of urine or serum for exposure assessment critically determines the concentrations reported in the studies, this information has been added in the discussion section. The revised manuscript now reads:
“In addition, taken together, the great heterogeneity in terms of study population (i.e. healthy adult subjects, healthy boys and girls, infant allergic rhinitis patients, arthritis rheumatoid patients, women diagnosed with PCOS, and diabetic and Crohn’s patients…), the applied methodological for both EDC (i.e. gas chromatography-mass spectrometry (GC-MS/MS), liquid chromatography-mass spectrometry (LC-MS/MS), isotope dilution-liquid chromatography-tandem mass spectrometry (ID-LC-MS/MS), enzyme-linked immunosorbent assay (ELISA)…) and inflammation assessment (i.e. ELISA, immunoturbidimetry, high performance liquid chromatography (HPLC)…), and the use of different biological matrices for exposure assessment (urine and serum) could explain, at least in part, the wide ranges reported in EDC and inflammatory biomarker levels shown in the different studies included in this review.”
Reviewer 2 Report
The work of Peinado et al. is an ambitious systematic review that tries to summarize the vast topic of bisphenols, parabens and benzophenones effects on human and environmental health with a focus on inflammation.
The title is concise and representative and so is the abstract.
Introduction:
Line 28-30: there are only two citation to support the affirmation, additional relevant citations should be added.
Line 31: " Bisphenols are..." state how many types of bisphenols are, their structural role in plastic composition as well as %, their prevalence and their analogues [1,2].
Line 57-59: " In addition, it has been postulated that inflammation might act as an alternative or complementary mechanism of action to the endocrine disruption hypothesis" this is too briefly stated, elaborate.
Line 68-60: "The regulation of inflammatory responses is complex and involves many different cell types (immune, epithelial, endothelial, and mesenchymal cells)" state citations.
Line 69-71: " Sometimes, the inflammatory response may not be properly regulated due to an inappropriate, misdirected, or excessive response from innate immune cells" elaborate and state citations.
The introduction chapter is severely underdeveloped.
Materials and Methods
Is there a reason why not all bisphenols have been included in the survey? including two that are identified as of very high concern [3,4].
The manuscript has potential value but one or even two major revisions are required prior to acceptance.
References
Daxi Liu, Pengfei Wu, Nan Zhao, Saisai Nie, Jiansheng Cui, Meirong Zhao, Hangbiao Jin, Differences of bisphenol analogue concentrations in indoor dust between rural and urban areas, Chemosphere,Volume 276,2021,130016,ISSN 0045- 6535,https://doi.org/10.1016/j.chemosphere.2021.130016.
2. Thoene, M., Dzika, E., Gonkowski, S., & Wojtkiewicz, J. (2020). Bisphenol S in Food Causes Hormonal and Obesogenic Effects Comparable to or Worse than Bisphenol A: A Literature Review. Nutrients, 12(2), 532. https://doi.org/10.3390/nu12020532
3. Kim, J. J., Kumar, S., Kumar, V., Lee, Y. M., Kim, Y. S., & Kumar, V. (2019). Bisphenols as a Legacy Pollutant, and Their Effects on Organ Vulnerability. International journal of environmental research and public health, 17(1), 112. https://doi.org/10.3390/ijerph17010112
4. https://echa.europa.eu/ro/-/group-assessment-of-bisphenols-identifies-need-for-restriction
Author Response
Reviewer #2:
The work of Peinado et al. is an ambitious systematic review that tries to summarize the vast topic of bisphenols, parabens and benzophenones effects on human and environmental health with a focus on inflammation.
The title is concise and representative and so is the abstract.
Response: We are grateful to this reviewer for the positive evaluation, whose comments have been addressed below:
Introduction:
Line 28-30: there are only two citation to support the affirmation, additional relevant citations should be added.
Response: As requested, the following additional relevant citations have been added:
Sifakis, S.; Androutsopoulos, V. P.; Tsatsakis, A. M.; Spandidos, D. A., Human exposure to endocrine disrupting chemicals: effects on the male and female reproductive systems. Environmental toxicology and pharmacology 2017, 51, 56-70.
Liu, W.; Zhou, Y.; Li, J.; Sun, X.; Liu, H.; Jiang, Y.; Peng, Y.; Zhao, H.; Xia, W.; Li, Y.; Cai, Z.; Xu, S., Parabens exposure in early pregnancy and gestational diabetes mellitus. Environment international 2019, 126, 468-475.
Engdahl, E.; Rüegg, J., Prenatal Exposure to Endocrine Disrupting Chemicals and Their Effect on Health Later in Life. In Beyond Our Genes: Pathophysiology of Gene and Environment Interaction and Epigenetic Inheritance, Teperino, R., Ed. Springer International Publishing: Cham, 2020; pp 53-77.
Huo, W.; Cai, P.; Chen, M.; Li, H.; Tang, J.; Xu, C.; Zhu, D.; Tang, W.; Xia, Y., The relationship between prenatal exposure to BP-3 and Hirschsprung's disease. Chemosphere 2016, 144, 1091-7.
Line 31: "Bisphenols are..." state how many types of bisphenols are, their structural role in plastic composition as well as %, their prevalence and their analogues [1,2].
Response: As requested, we have included more relevant information on bisphenols in the second paragraph of the introduction section, including the references suggested by the reviewer, among others. The revised paragraph reads as follows:
“Bisphenols are non-persistent phenolic compounds widely used in the synthesis of polycarbonate plastics and epoxy resins, and are frequently found in the linings of canned and packaged food containers, thermal receipts, and dental sealants [7-8]. Bisphenol A (BPA) is the most studied congener and is one of the most produced chemicals in the world [9], reaching a global production volume of more than 5 million tons [10], and with an annual growth rate that reached 4.6% between 2013 and 2019 [11]. Moreover, data from biomonitoring studies indicate that BPA exposure is ubiquitous and widespread in the population, with BPA concentrations found in 90.0% of the general population in industrialized countries [12-13]. Due to the harmful effects inherent to exposure to BPA, some international government regulators have banned its use in baby bottles and cosmetics [14]. As an alternative to BPA, bisphenol analogues structurally similar to BPA began to be produced, such as bisphenol S, bisphenol F, bisphenol AF, tetrabromobisphenol, bisphenol A-glycidyl methacrylate, bisphenol A diglycidyl ether and bisphenol F diglycidyl ether [15-16]. However, previous evidence has suggested that these analogues may be even more harmful than the original BPA in some situations [16].”
A similar approach has been made for paraben and benzophenones. The revised version of the manuscript now reads:
“The family of PBs includes alkyl esters of p-hydroxybenzoic acid and is used in a wide range of cosmetics and personal care products (PCPs) as well as in food packaging due to their antimicrobial and preservative properties [17-21]. The main congeners of PBs are methylparaben, ethylparaben, propylparaben, and butylparaben. BPs are aromatic ketones included in a wide variety of cosmetics, PCPs and textiles due to their properties as UV filters [22-23], and include different congeners, such as benzophenone 1, benzophenone 2, benzophenone 3, 4-hydroxybenzophenone, benzophenone 6, and benzophenone 8.”
Line 57-59: " In addition, it has been postulated that inflammation might act as an alternative or complementary mechanism of action to the endocrine disruption hypothesis" this is too briefly stated, elaborate.
Response: As indicated by the reviewer, we have added more information on this topic in the introduction section. The revised paragraph now reads as follows:
“In addition, it has been postulated that inflammation might act as an alternative or complementary mechanism of action to the endocrine disruption hypothesis, given that they could promote an inflammatory milieu through activation of ERα nuclear receptors [41-43]. In this sense, previous evidence has reported the presence of estrogen-dependent nuclear receptors in promoter regions of genes related to the inflammatory response, such as ERα and ERβ [41, 44-46], suggesting that the origin and development of an inflammatory response could be an indirect consequence of endocrine alterations promoted by these compounds with hormonal activity.”
Line 68-60: "The regulation of inflammatory responses is complex and involves many different cell types (immune, epithelial, endothelial, and mesenchymal cells)" state citations.
Response: As requested, we have state the following citations supporting the previously mentioned sentence:
Dietert, R. R., Misregulated inflammation as an outcome of early-life exposure to endocrine-disrupting chemicals. Reviews on environmental health 2012, 27 (2-3), 117-31.
Khan, D.; Ansar Ahmed, S., The Immune System Is a Natural Target for Estrogen Action: Opposing Effects of Estrogen in Two Prototypical Autoimmune Diseases. Frontiers in immunology 2015, 6, 635.
Line 69-71: "Sometimes, the inflammatory response may not be properly regulated due to an inappropriate, misdirected, or excessive response from innate immune cells" elaborate and state citations.
Response: As requested, we have elaborated and expanded this paragraph in the introduction section, including additional relevant information and stating the corresponding citations. The revised paragraph now reads as follows:
“Like most immune responses, the inflammatory phenomenon is tightly regulated, and a proper and precise balance between proinflammatory and anti-inflammatory immune responses is required to effectively eliminate infectious pathogens while limiting immune damage in the host [48]. The regulation of inflammatory responses is complex, involves many different cell types (immune, epithelial, endothelial, and mesenchymal cells) [41, 47], and sometimes it may not be properly regulated. A misregulated inflammation could be originated when the response among innate immune cells is inappropriate for the type of defense needed against the invader, the response is misdirected based on the location of the strange agent, the response is overproduced, and/or the response is not beneficially resolved for the host [41, 47]. Deviations from tightly regulated inflammation present a significant health risk because unresolved inflammation can compromise tissue function and increase the risk of several chronic cardiovascular diseases and metabolic disorders [47].”
The introduction chapter is severely underdeveloped.
Response: So that the introduction chapter is not underdeveloped, relevant information on the description of bisphenols, parabens and benzophenones has been added. In addition, information regarding the inflammatory process has also been included, considering the key role that it could play as a possible complementary mechanism of action of these compounds and the regulation of the inflammatory response.
Materials and Methods
Is there a reason why not all bisphenols have been included in the survey? including two that are identified as of very high concern [3,4].
Response: The aim of this systematic review was to include all bisphenol congeners. Therefore, the term “bisphenols” was included in the search strategy (Supplementary Table S1). Moreover, during screening and full-text revision process, those studies assessing exposure to any bisphenol congener were included. In addition, some specific terms from those most studied congeners were also included in the search strategy in order to reduce possible selection biases. However, some bisphenols without specific terms were included such as bisphenol AF (i.e. study no. 16). Nevertheless, after reviewing the references mentioned by the reviewer, we have carried out a review of the literature considering the bisphenol congeners without a specific term in our search strategy and identified as of very high concern. The results obtained did not show any additional article that met the inclusion criteria to be included in this systematic review.
The manuscript has potential value but one or even two major revisions are required prior to acceptance.
References
- Daxi Liu, Pengfei Wu, Nan Zhao, Saisai Nie, Jiansheng Cui, Meirong Zhao, Hangbiao Jin, Differences of bisphenol analogue concentrations in indoor dust between rural and urban areas, Chemosphere,Volume 276,2021,130016,ISSN 0045- 6535,https://doi.org/10.1016/j.chemosphere.2021.130016.
- Thoene, M., Dzika, E., Gonkowski, S., & Wojtkiewicz, J. (2020). Bisphenol S in Food Causes Hormonal and Obesogenic Effects Comparable to or Worse than Bisphenol A: A Literature Review. Nutrients, 12(2), 532. https://doi.org/10.3390/nu12020532
- Kim, J. J., Kumar, S., Kumar, V., Lee, Y. M., Kim, Y. S., & Kumar, V. (2019). Bisphenols as a Legacy Pollutant, and Their Effects on Organ Vulnerability. International journal of environmental research and public health, 17(1), 112. https://doi.org/10.3390/ijerph17010112
- https://echa.europa.eu/ro/-/group-assessment-of-bisphenols-identifies-need-for-restriction
Reviewer 3 Report
The aim of the study was to summarize the current evidence on the relationship between human exposure to of bisphenols, parabens and benzophenones and levels of inflammatory biomarkers. The authors conducted the systematic review of peer-reviewed original papers published until February 2023. A total of 20 papers met inclusion/exclusion criteria. Most of the studies reported significant associations between any of selected chemicals and some pro-inflammatory biomarkers. The systematic review has identified consistent positive association between PBS and/or BPs and inflammation. The authors concluded that more studies is needed to get a better understanding on mechanisms of action underlying bisphenols, parabens and benzophenones and the critical role that the inflammation could play.
The paper is prepared correctly. The results obtained are interesting and add more information to the science about the health effects of bisphenols, parabens and benzophenones.
The manuscript is recommended for publication after minor correction.
The main shortcomings are Tables. The letters in Tables 2, 3 i 4 are very small, so the contents is hard to read. They should be rearranged and enlarged.
Author Response
Reviewer #3:
The aim of the study was to summarize the current evidence on the relationship between human exposure to of bisphenols, parabens and benzophenones and levels of inflammatory biomarkers. The authors conducted the systematic review of peer-reviewed original papers published until February 2023. A total of 20 papers met inclusion/exclusion criteria. Most of the studies reported significant associations between any of selected chemicals and some pro-inflammatory biomarkers. The systematic review has identified consistent positive association between PBS and/or BPs and inflammation. The authors concluded that more studies is needed to get a better understanding on mechanisms of action underlying bisphenols, parabens and benzophenones and the critical role that the inflammation could play.
The paper is prepared correctly. The results obtained are interesting and add more information to the science about the health effects of bisphenols, parabens and benzophenones.
The manuscript is recommended for publication after minor correction.
Response: We are grateful for the positive evaluation by this reviewer, whose concern has been addressed below.
The main shortcomings are Tables. The letters in Tables 2, 3 i 4 are very small, so the contents is hard to read. They should be rearranged and enlarged.
Response: As requested, we have modified the format of tables 2, 3 and 4, which have been rearranged and enlarged to make the content easier to read.
Round 2
Reviewer 2 Report
The authors have fully covered the requirements. I have no further remarks. My decision is: Accept in current form.